# Identifying and Controlling Important Neurons in Neural Machine Translation

**Anthony Bau**[1*]      **Yonatan Belinkov**[1*]      **Hassan Sajjad**[2]
**Nadir Durrani**[2]      **Fahim Dalvi**[2]      **James Glass**[1]

[1]MIT Computer Science and Artificial Intelligence Laboratory, Cambridge, MA 02139, USA
[2]Qatar Computing Research Institute, HBKU Research Complex, Doha 5825, Qatar
`{abau,belinkov,glass}@mit.edu`
`{hsajjad,ndurrani,faimaduddin}@qf.org.qa`

## Abstract

Neural machine translation (NMT) models learn representations containing substantial linguistic information. However, it is not clear if such information is fully distributed or if some of it can be attributed to individual neurons. We develop unsupervised methods for discovering important neurons in NMT models. Our methods rely on the intuition that different models learn similar properties, and do not require any costly external supervision. We show experimentally that translation quality depends on the discovered neurons, and find that many of them capture common linguistic phenomena. Finally, we show how to control NMT translations in predictable ways, by modifying activations of individual neurons.

## 1 Introduction

Neural machine translation (NMT) systems achieve state-of-the-art results by learning from large amounts of example translations, typically without additional linguistic information. Recent studies have shown that representations learned by NMT models contain a non-trivial amount of linguistic information on multiple levels: morphological (Belinkov et al., 2017a; Dalvi et al., 2017), syntactic (Shi et al., 2016b), and semantic (Hill et al., 2017). These studies use trained NMT models to generate feature representations for words, and use these representations to predict certain linguistic properties. This approach has two main limitations. First, it targets the whole vector representation and fails to analyze individual dimensions in the vector space. In contrast, previous work found meaningful individual neurons in computer vision (Zeiler & Fergus, 2014; Zhou et al., 2016; Bau et al., 2017, among others) and in a few NLP tasks (Karpathy et al., 2015; Radford et al., 2017; Qian et al., 2016a). Second, these methods require external supervision in the form of linguistic annotations. They are therefore limited by available annotated data and tools.

In this work, we make initial progress towards addressing these limitations by developing unsupervised methods for analyzing the contribution of *individual neurons* to NMT models. We aim to answer the following questions:

- How important are individual neurons for obtaining high-quality translations?
- Do individual neurons in NMT models contain interpretable linguistic information?
- Can we control MT output by intervening in the representation at the individual neuron level?

To answer these questions, we develop several unsupervised methods for ranking neurons according to their importance to an NMT model. Inspired by work in machine vision (Li et al., 2016b), we hypothesize that different NMT models learn similar properties, and therefore similar important neurons should emerge in different models. To test this hypothesis, we map neurons between pairs of trained NMT models using several methods: correlation analysis, regression analysis, and SVCCA, a recent method combining singular vectors and canonical correlation analysis (Raghu et al., 2017). Our mappings yield lists of candidate neurons containing shared information across models. We

---

*Equal contribution

then evaluate whether these neurons carry important information to the NMT model by masking their activations during testing. We find that highly-shared neurons impact translation quality much more than unshared neurons, affirming our hypothesis that *shared information matters*.

Given the list of important neurons, we then investigate what linguistic properties they capture, both qualitatively by visualizing neuron activations and quantitatively by performing supervised classification experiments. We were able to identify neurons corresponding to several linguistic phenomena, including morphological and syntactic properties.

Finally, we test whether intervening in the representation at the individual neuron level can help *control the translation*. We demonstrate the ability to control NMT translations on three linguistic properties—tense, number, and gender—to varying degrees of success. This sets the ground for controlling NMT in desirable ways, potentially reducing system bias to properties like gender.

Our work indicates that not all information is distributed in NMT models, and that many human-interpretable grammatical and structural properties are captured by individual neurons. Moreover, modifying the activations of individual neurons allows controlling the translation output according to specified linguistic properties. The methods we develop here are task-independent and can be used for analyzing neural networks in other tasks. More broadly, our work contributes to the localist/distributed debate in artificial intelligence and cognitive science (Gayler & Levy, 2011) by investigating the important case of neural machine translation.

## 2 RELATED WORK

Much recent work has been concerned with analyzing neural representations of linguistic units, such as word embeddings (Köhn, 2015; Qian et al., 2016b), sentence embeddings (Adi et al., 2016; Ganesh et al., 2017; Brunner et al., 2018), and NMT representations at different linguistic levels: morphological (Belinkov et al., 2017a), syntactic (Shi et al., 2016b), and semantic (Hill et al., 2017; Belinkov et al., 2017b). These studies follow a common methodology of evaluating learned representations on external supervision by training classifiers or measuring other kinds of correlations (Belinkov & Glass, 2019). Thus they are limited to the available supervised annotation. In addition, these studies do not typically consider individual dimensions. In contrast, we propose intrinsic unsupervised methods for detecting important neurons based on correlations between independently trained models. A similar approach was used to analyze vision models (Li et al., 2016b), but to the best of our knowledge these ideas were not applied to NMT or other NLP models before.

In computer vision, individual neurons were shown to capture meaningful information (Zeiler & Fergus, 2014; Zhou et al., 2016; Bau et al., 2017). Even though some doubts were cast on the importance of individual units (Morcos et al., 2018), recent work stressed their contribution to predicting specific object classes via masking experiments similar to ours (Zhou et al., 2018). A few studies analyzed individual neurons in NLP. For instance, neural language models learn specific neurons that activate on brackets (Karpathy et al., 2015), sentiment (Radford et al., 2017), and length (Qian et al., 2016a). Length-specific neurons were also found in NMT (Shi et al., 2016a), but generally not much work has been devoted to analyzing individual neurons in NMT. We aim to address this gap.

## 3 METHODOLOGY

Much recent work on analyzing NMT relies on supervised learning, where NMT representations are used as features for predicting linguistic annotations (see Section 2). However, such annotations may not be available, or may constrain the analysis to a particular scheme.

Instead, we propose to use different kinds of correlations between neurons from different models as a measure of their importance. Suppose we have $M$ such models and let $\mathbf{h}_t^m[i]$ denote the activation of the $i$-th neuron in the encoder of the $m$-th model for the $t$-th word.[1] These may be models from different training epochs, trained with different random initializations or datasets, or even different architectures—all realistic scenarios that researchers often experiment with. Let $\mathbf{x}_i^m$ denote a random variable corresponding to the $i$-th neuron in the $m$-th model. $\mathbf{x}_i^m$ maps words to their neuron activations: $\mathbf{x}_i^m : t \mapsto \mathbf{h}_t^m[i]$. Similarly, let $\mathbf{x}^m$ denote a random vector corresponding to the activations of all neurons in the $m$-th model: $\mathbf{x}^m : t \mapsto \boldsymbol{h}_t^m$.

---

[1] We only consider neurons from the top layer, although the approach can also be applied to other layers.

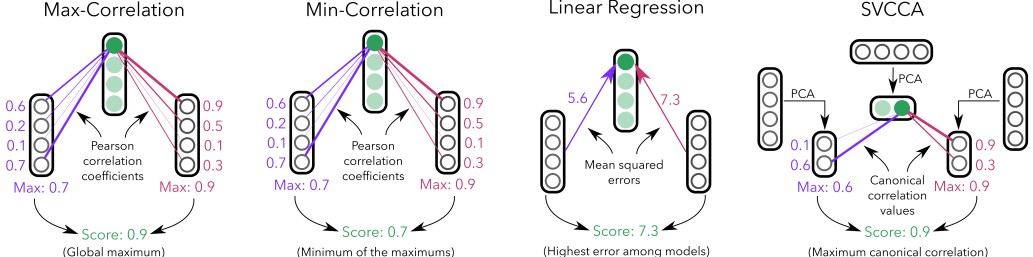

Figure 1: An illustration of the correlation methods, showing how to compute the score for one neuron using each of the methods. Here the number of models is $M = 3$, each having four neurons.

We consider four methods for *ranking* neurons, based on correlations between pairs of models. Our hypothesis is that different NMT models learn similar properties, and therefore similar important neurons emerge in different models, akin to neural vision models (Li et al., 2016b). Our methods capture different levels of localization/distributivity, as described next. See Figure 1 for illustration.

### 3.1 UNSUPERVISED CORRELATION METHODS

**Maximum correlation**    The maximum correlation (`MaxCorr`) of neuron $x_i^m$ looks for the highest correlation with any neuron in all other models:

$$\texttt{MaxCorr}(x_i^m) = \max_{j, m' \neq m} |\rho(x_i^m, x_j^{m'})|$$ (1)

where $\rho(x, y)$ is the Pearson correlation coefficient between x and y. We then rank the neurons in model $m$ according to their `MaxCorr` score. We repeat this procedure for every model $m$. This score looks for neurons that capture properties that emerge strongly in two separate models.

**Minimum correlation**    The minimum correlation (`MinCorr`) of neuron $x_i^m$ looks for the neurons most correlated with $X_i^m$ in each of the other models, but selects the one with the lowest correlation:

$$\texttt{MinCorr}(x_i^m) = \min_{m' \neq m} \max_j |\rho(x_i^m, x_j^{m'})|$$ (2)

Neurons in model $m$ are ranked according to their `MinCorr` score. This tries to find neurons that are well correlated with many other models, even if they are not the overall most correlated ones.

**Regression ranking**    We perform linear regression (`LinReg`) from the full representation of another model $\mathbf{x}^{m'}$ to the neuron $x_i^m$. Then we rank neurons by the regression mean squared error. This attempts to find neurons whose information might be distributed in other models.

**SVCCA**    Singular vector canonical correlation analysis (`SVCCA`) is a recent method for analyzing neural networks (Raghu et al., 2017). In our implementation, we perform PCA on each model's representations $\mathbf{x}^m$ and take enough dimensions to account for 99% of the variance. For each pair of models, we obtain the canonically correlated basis, and rank the basis directions by their CCA coefficients. This attempts to capture information that may be distributed in less dimensions than the whole representation. In this case we get a ranking of directions, rather than individual neurons.

### 3.2 VERIFYING DETECTED NEURONS

We want to verify that neurons ranked highly by the unsupervised methods are indeed important for the NMT models. We consider quantitative and qualitative techniques for verifying their importance.

**Erasing Neurons**    We test importance of neurons by erasing some of them during translation. Erasure is a useful technique for analyzing neural networks (Li et al., 2016a). Given a ranked list of neurons $\pi$, where $\pi(i)$ is the rank of neuron $x_i$, we zero-out increasingly more neurons according to the ranking $\pi$, starting from either the top or the bottom of the list. Our hypothesis is that erasing neurons from the top would hurt translation performance more than erasing from the bottom.

Concretely, we first run the entire encoder as usual, then zero out specific neurons from all source hidden states $\{\mathbf{h}_1, \dots, \mathbf{h}_n\}$ before running the decoder. For `MaxCorr`, `MinCorr`, and `LinReg`,

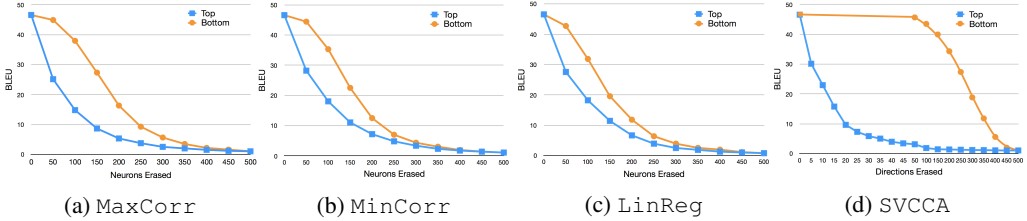


(a) MaxCorr     (b) MinCorr     (c) LinReg     (d) SVCCA


Figure 2: Erasing neurons (or SVCCA directions) from the top and bottom of the list of most important neurons (directions) ranked by different unsupervised methods, in an English-Spanish model.

we zero out individual neurons. To erase $k$ directions found by SVCCA, we instead project the embedding $\boldsymbol{E}$ (corresponding to all activations of a given model over a dataset) onto the space spanned by the non-erased directions: $\boldsymbol{E}' = \boldsymbol{E}(\boldsymbol{C}(\boldsymbol{C}^T\boldsymbol{C})^{-1}\boldsymbol{C}^T)$, where $\boldsymbol{C}$ is the CCA projection matrix with the first or last $k$ columns removed. This corresponds to erasing from the top or bottom.

**Supervised Verification**  While our focus is on unsupervised methods for finding important neurons, we also utilize supervision to verify our results. Importantly, these experiments are done post-hoc, after having a candidate neuron to examine. Since training a supervised classifier on every neuron is costly, we instead report simple metrics that can be easily computed. Specifically, we sometimes report the expected conditional variance of neuron activations conditioned on some property. In other cases we found it useful to estimate a Gaussian mixture model (GMM) for predicting a label and measure its prediction quality. The number of mixtures in the GMM is set according to the number of classes in the predicted property (e.g. 2 mixtures when predicting tokens inside or outside of parentheses), and its parameters are estimated using the mean and variance of the neuron activation conditioned on each class. We obtain linguistic annotations with Spacy: spacy.io.

**Visualization**  Interpretability of machine learning models remains elusive (Lipton, 2016), but visualizing can be an instructive technique. Similar to previous work analyzing neural networks in NLP (Elman, 1991; Karpathy et al., 2015; Kádár et al., 2016), we visualize activations of neurons and observe interpretable behavior. We will illustrate this with example heatmaps below.

## 4 EXPERIMENTAL SETUP

**Data**  We use the United Nations (UN) parallel corpus (Ziemski et al., 2016) for all experiments. We train models from English to 5 languages: Arabic, Chinese, French, Russian, and Spanish, as well as an English-English auto-encoder. For each target language, we train 3 models on different parts of the training set, each with 500K sentences. In total, we have 18 models. This setting allows us to compare models trained on the same language pairs but different training data, as well as models trained on different language pairs. We evaluate on the official test set.[2]

**MT training**  We train 500 dimensional 2-layer LSTM encoder-decoder models with attention (Bahdanau et al., 2014). In order to study both word and sub-word properties, we use a word representation based on a character convolutional neural network (charCNN) as input to both encoder and decoder, which was shown to learn morphology in language modeling and NMT (Kim et al., 2015; Belinkov et al., 2017a).[3] While we focus here on recurrent NMT, our approach can be applied to other models like the Transformer (Vaswani et al., 2017), which we leave for future work.

## 5 RESULTS

### 5.1 ERASURE EXPERIMENTS

Figure 2 shows erasure results using the methods from Section 3.1, on an English-Spanish model. For all four methods, erasing from the top hurts performance much more than erasing from the

---

[2]Our experimental evaluation is focused on models trained on different parts of the training data, but we provide a short discussion of results on models from different epochs of a given training run in Appendix A.4.

[3]We used this representation rather than BPE sub-word units (Sennrich et al., 2016) to facilitate interpretability with respect to specific words. In the experiments, we report word-based results unless noted otherwise.

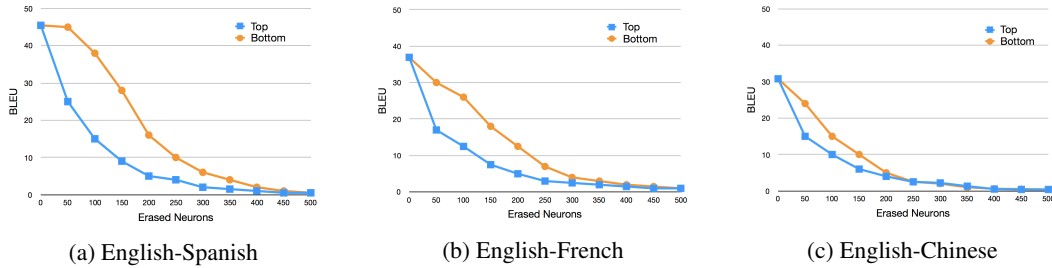

|                     |                   |                    |
|:-------------------:|:-----------------:|:------------------:|
| (a) English-Spanish | (b) English-French | (c) English-Chinese |

Figure 3: Erasing neurons from the top or bottom of the `MaxCorr` ranking in three language pairs.

Table 1: Top 10 neurons (or `SVCCA` directions) in an English-Spanish model according to the four methods, and the percentage of explained variance by conditioning on position or token identity.

| | MaxCorr | | | MinCorr | | | LinReg | | | SVCCA | |
|-----|---------|------|-----|---------|------|-----|--------|------|------|-----|------|
| ID  | Pos     | Tok  | ID  | Pos     | Tok  | ID  | Pos    | Tok  | | Pos | Tok  |
| 464 | **92%** | 10%  | 342 | **88%** | 7.9% | 464 | **92%**| 10%  | | 86% | 26%  |
| 342 | **88%** | 7.9% | 464 | **92%** | 10%  | 260 | 0.71%  | **94%**| | 1.6% | **90%** |
| 260 | 0.71%   | **94%** | 260 | 0.71% | **94%** | 139 | 0.86% | **93%** | | 7.5% | **85%** |
| 49  | 11%     | 6.1% | 383 | **67%** | 6.5% | 494 | 3.5%   | **96%**| | 20% | **79%** |
| 124 | **77%** | 48%  | 250 | **63%** | 6.8% | 342 | **88%**| 7.9% | | 1.1% | **89%** |
| 394 | 0.38%   | 22%  | 124 | **77%** | 47%  | 228 | 0.38%  | **96%**| | 10% | **76%** |
| 228 | 0.38%   | **96%** | 485 | **64%** | 10% | 317 | 1.5%   | **83%**| | 30% | **57%** |
| 133 | 0.14%   | **87%** | 480 | **70%** | 12% | 367 | 0.44%  | **89%**| | 24% | **55%** |
| 221 | 1%      | 30%  | 154 | **63%** | 15%  | 106 | 0.25%  | **92%**| | 23% | **60%** |
| 90  | 0.49%   | 28%  | 139 | 0.86%   | **93%** | 383 | **67%** | 6.5% | | 18% | **63%** |

bottom. This confirms our hypothesis that neurons ranked higher by our methods have a larger impact on translation quality. Comparing erasure with different rankings, we find similar patterns with `MaxCorr`, `MinCorr`, and `LinReg`: erasing the top ranked 10% (50 neurons) degrades BLEU by 15-20 points, while erasing the bottom 10% neurons only hurts by 2-3 points. In contrast, erasing `SVCCA` directions results in rapid degradation: 15 BLEU point drop when erasing 1% (5) of the top directions, and poor performance when erasing 10% (50). This indicates that top `SVCCA` directions capture very important information in the model. We analyze these top neurons and directions in the next section, finding that top `SVCCA` directions focus mostly on identifying specific words.

Figure 3 shows the results of `MaxCorr` when erasing neurons from top and bottom, using models trained on three language pairs. In all cases, erasing from the top hurts performance more than erasing from the bottom. We found similar trends with other language pairs and ranking methods.

## 5.2 EVALUATING TOP NEURONS

What kind of information is captured by the neurons ranked highly by each of our ranking methods? Previous work found specific neurons in NMT that capture position of words in the sentence (Shi et al., 2016a). Do our methods capture similar properties? Indeed, we found that many top neurons capture position. For instance, Table 1 shows the top 10 ranked neurons from an English-Spanish model according to each of the methods. The table shows the percent of variance in neuron activation that is eliminated by conditioning on position in the sentence, calculated over the test set. Similarly, it shows the percent of explained variance by conditioning on the current token identity.

We observe an interesting difference between the ranking methods. `LinReg` and especially `SVCCA`, which are both computed by using multiple neurons, tend to find information determined by the identity of the current token. `MinCorr` and to a lesser extent `MaxCorr` tend to find position information. This suggests that information about the current token is often distributed in multiple neurons, which can be explained by the fact that tokens carry multiple kinds of linguistic information. In contrast, position is a fairly simple property that the NMT encoder can represent in a small number of neurons. That fact that many top `MinCorr` neurons capture position suggests that this kind of information is captured in multiple models in a similar way.

Table 2: $F_1$ scores of the top two neurons from each network for detecting tokens inside parentheses, and the ranks of the top neuron according to our intrinsic unsupervised methods.

| Neuron | 1st | 2nd | Max | Min | Reg | Neuron | 1st | 2nd | Max | Min | Reg |
|---|---|---|---|---|---|---|---|---|---|---|---|
| en-es-1:232 | 0.66 | 0.23 | 15 | 45 | 27 | en-ar-3:331 | 0.76 | 0.29 | 18 | 93 | 50 |
| en-es-2:208 | 0.69 | 0.21 | 9 | 44 | 22 | en-ru-1:259 | 0.74 | 0.23 | 11 | 48 | 45 |
| en-es-3:47 | 0.56 | 0.22 | 12 | 35 | 24 | en-ru-2:23 | 0.81 | 0.15 | 11 | 72 | 32 |
| en-fr-1:449 | 0.51 | 0.24 | 38 | 43 | 15 | en-ru-3:214 | 0.74 | 0.35 | 25 | 67 | 116 |
| en-fr-2:361 | 0.58 | 0.25 | 29 | 45 | 61 | en-zh-1:49 | 0.68 | 0.53 | 6 | 84 | 64 |
| en-fr-3:205 | 0.35 | 0.33 | 228 | 236 | 206 | en-zh-2:159 | 0.75 | 0.49 | 6 | 48 | 38 |
| en-ar-1:212 | 0.45 | 0.41 | 85 | 79 | 41 | en-zh-3:467 | 0.53 | 0.31 | 6 | 60 | 48 |
| en-ar-2:166 | 0.75 | 0.26 | 6 | 119 | 67 | | | | | | |

## 5.3 LINGUISTICALLY INTERPRETABLE NEURONS

Neurons that activate on specific tokens or capture position in the sentence are important in some of the methods, as shown in the previous section. But they are not highly ranked in all methods and are also less interesting from the perspective of capturing language information. In this section, we investigate several linguistic properties by measuring predictive capacity and visualizing neuron activations. The supplementary material discusses more properties. Further analysis of linguistically interpretable neurons is available in Dalvi et al. (2019a).

**Parentheses** Table 2 shows top neurons from each model for predicting that tokens are inside/outside of parentheses, quotes, or brackets, estimated by a GMM model. Often, the parentheses neuron is unique (low scores for the 2nd best neuron), suggesting that this property tends to be relatively localized. Generally, neurons that detect parentheses were ranked highly in most models by the `MaxCorr` method, indicating that they capture important patterns in multiple networks.

Figure 4 visualizes the most predictive neuron in an English-Spanish model. It activates positively (red) inside parentheses and negatively (blue) outside. Similar neurons were found in RNN language models (Karpathy et al., 2015). Next we consider more complicated linguistic properties.

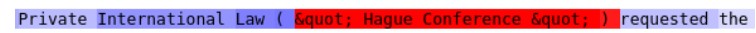

Figure 4: Visualization of a parentheses neuron from an English-Spanish model.

**Tense** We annotated the test data for verb tense (with Spacy) and trained a GMM model to predict tense from neuron activations. Figure 5 shows activations of a top-scoring neuron (0.66 $F_1$) from the English-Arabic model on the first 5 test sentences. It tends to activate positively (red color) on present tense ("recognizes", "recalls", "commemorate") and negatively (blue color) on past tense ("published", "disbursed", "held"). These results are obtained with a charCNN representation, which is sensitive to common suffixes like "-ed", "-es". However, this neuron also detects irregular past tense verbs like "held", suggesting that it captures context in addition to sub-word information. The neuron also makes some mistakes by activating weakly positively on nouns ending with "s" ("videos", "punishments"), presumably because it gets confused with the 3rd person present tense. Similarly, it activates positively on "Spreads", even though it functions as a noun in this context.

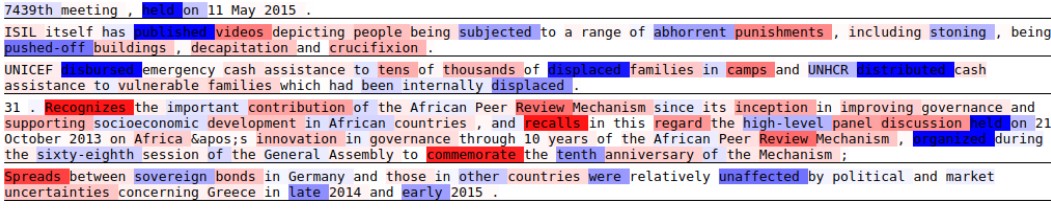

Figure 5: Visualization of a neuron from an English-Arabic model that activates on verb tense: negative/positive for past/present. Examples shown are the first 5 sentences in the test set.

Table 3: Correlations ($\rho$) and tense predictiveness ($F_1$ score and Rank) of the most correlated neurons in all models relative to a tense neuron in an English-Arabic model.

| Neuron | $\rho$ | $F_1$ | R | Neuron | $\rho$ | $F_1$ | R | Neuron | $\rho$ | $F_1$ | R |
|---|---|---|---|---|---|---|---|---|---|---|---|
| en-ar-1:8 | 1 | 0.67 | 1 | en-fr-1:314 | 0.48 | 0.46 | 1 | en-zh-1:383 | -0.51 | 0.54 | 1 |
| en-ar-2:303 | 0.57 | 0.43 | 3 | en-fr-2:333 | -0.58 | 0.61 | 2 | en-zh-2:451 | -0.18 | 0.29 | 2 |
| en-ar-3:108 | 0.66 | 0.62 | 1 | en-fr-3:399 | -0.69 | 0.6 | 1 | en-zh-3:165 | -0.30 | 0.28 | 6 |
| en-es-1:114 | 0.56 | 0.59 | 1 | en-ru-1:330 | -0.39 | 0.54 | 1 | en-en-1:180 | -0.03 | 0 | 471 |
| en-es-2:282 | -0.36 | 0.46 | 3 | en-ru-2:489 | 0.29 | 0.08 | 27 | en-en-2:24 | -0.19 | 0 | 319 |
| en-es-3:255 | -0.22 | 0.26 | 8 | en-ru-3:397 | -0.50 | 0.52 | 1 | en-en-3:486 | -0.33 | 0.51 | 1 |

Table 3 shows correlations of neurons most correlated with this tense neuron, according to `MaxCorr`. All these neurons are highly predictive of tense: all but 3 are in the top 10 and 8 out of 15 (non-auto-encoder) neurons have the highest $F_1$ score for predicting tense. The auto-encoder English models are an exception, exhibiting much lower correlations with the English-Arabic tense neuron. This suggests that tense emerges in a "real" NMT model, but not in an auto-encoder that only learns to copy. Interestingly, English-Chinese models have somewhat lower correlated neurons with the tense neuron, possibly due to the lack of explicit tense marking in Chinese. The encoder does not need to pay as much attention to tense when generating representations for the decoder.

**Other Properties** We found many more linguistic properties by visualizing top neurons ranked by our methods, especially with `MaxCorr`. We briefly mention some of these here and provide more details and quantitative results in the appendix. We found neurons that activate on numbers, dates, adjectives, plural nouns, auxiliary verbs, and more. We also investigated noun phrase segmentation, a compositional property above the word level, and found high-scoring neurons (60-80% accuracy) in every network. Many of these neurons were ranked highly by the `MaxCorr` method. In contrast, other methods did not rank such neurons very highly. See Table 5 in the appendix for the full results.

Some neurons have quite complicated behavior. For example, when visualizing neurons highly ranked by `MaxCorr` we found a neuron that activates on numbers in the beginning of a sentence, but not elsewhere (see Figure 9 in the appendix). It would be difficult to conceive of a supervised prediction task which would capture this behavior a-priori, without knowing what to look for. Our unsupervised methods are flexible enough to find any neurons deemed important by the NMT model, without constraining the analysis to properties for which we have supervised annotations.

## 6 Controlling Translations

In this section, we explore a potential benefit of finding important neurons with linguistically meaningful properties: controlling the translation output. This may be important for mitigating biases in neural networks. For instance, gender stereotypes are often reflected in automatic translations, as the following motivating examples from Google Translate demonstrate.[4]

(1)  a. o bir doctor         (2)  a. o bir hemşire

   b. he is a doctor            b. she is a nurse

The Turkish sentences (1a, 2a) have no gender information—they can refer to either male or female. But the MT system is biased to think that doctors are usually men and nurses are usually women, so its generated translations (1b, 2b) represent these biases. If we know the correct gender from another source such as metadata, we may want to encourage the system to output a translation with the correct gender. We make here a modest step towards this goal by intervening in neuron activations to induce a desired translation.[5]

---

[4]Retrieved on 9/16/2018. This particular example has recently been addressed in Google Translate by providing alternative gendered translation. For more biased examples, see `mashable.com/2017/11/30/google-translate-sexism`.

[5]In concurrent work, Giulianelli et al. (2018) intervened in activations of a language model to improve its ability to capture verb number.

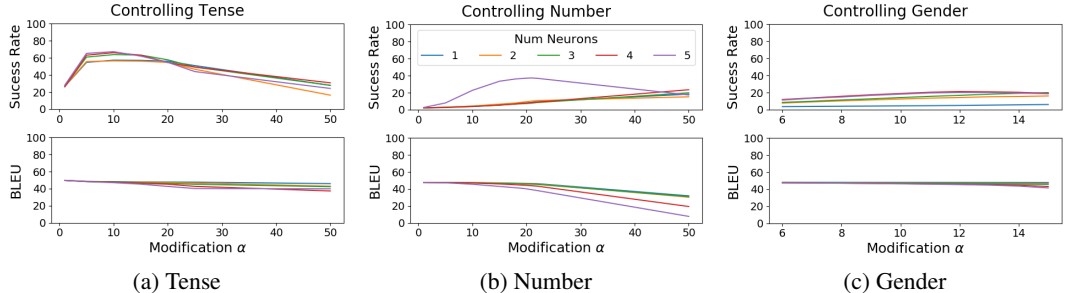

Figure 6: Success rates and BLEU scores for controlling NMT by modifying neuron activations.

We conjecture that if a given neuron matters to the model, then we can control the translation in predictable ways by modifying its activations. To do this, we first encode the source sentence as usual. Before decoding, we set the activation of a particular neuron in the encoder state to a value $\alpha$, which is a function of the mean activations over a particular property (defined below). To evaluate our ability to control the translation, we design the following protocol:

1. Tag the source and target sentences in the development set with a desired property, such as gender (masculine/feminine). We use Spacy for these tags.

2. Obtain word alignments for the development set using an alignment model trained on 2 million sentences of the UN data. We use `fast_align` (Dyer et al., 2013) with default settings.

3. For every neuron in the encoder, predict the target property on the word aligned to its source word activations using a supervised GMM model.[6]

4. For every word having a desired property, modify the source activations of the top $k$ neurons found in step 3 and generate a modified translation. The modification value is defined as $\alpha = \mu_1 + \beta(\mu_1 - \mu_2)$, where $\mu_1$ and $\mu_2$ are mean activations of the property we modify from and to, respectively (e.g. modifying gender from masculine to feminine), and $\beta$ is a hyper-parameter.

5. Tag the output translation and word-align it to the source. Declare *success* if the source word was aligned to a target word with the desired property value (e.g. feminine).

## 6.1 Results

Figure 6 shows translation control results in an English-Spanish model. We report success rate—the percentage of cases where the word was aligned to a target word with the desired property—and the effect on BLEU scores, when varying $\alpha$. Our tense control results are the most successful, with up to 67% success rate for changing past-to-present. Modifications generally degrade BLEU, but the loss at the best success rate is not large (2 BLEU points). Appendix A.2 provides more tense results.

Controlling other properties seems more difficult, with the best success rate for controlling number at 37%, using the top 5 number neurons. Gender is the most difficult to control, with a 21% success rate using the top 5 neurons. Modifying even more neurons did not help. We conjecture that these properties are more distributed than tense, which makes controlling them more difficult. Future work can explore more sophisticated methods for controlling multiple neurons simultaneously.

## 6.2 Example translations

We provide examples of controlling translation of number, gender, and tense. While these are cherry-picked examples, they illustrate that the controlling procedure can work in multiple properties and languages. We discuss language-specific patterns below.

---

[6]This is different from our results in the previous section, where we predicted a source-side property, because here we seek neurons that are predictive of target-side properties to facilitate controlling the translation.

Table 4: Examples for controlling translation by modifying activations of different neurons on the *italicized* source words. $\alpha$ = modification value (–, no modification).

(a) Controlling number when translating "The interested *parties*" to Spanish.

| $\alpha$ | Translation | Num | $\alpha$ | Translation | Num |
|---|---|---|---|---|---|
| -1 | abiertas particulares | pl. | 0.125 | La parte interesada | sing. |
| -0.5 | Observaciones interesadas | pl. | 0.25 | Cuestion interesada | sing. |
| -0.25, -0.125, 0 | Las partes interesadas | pl. | 0.5, 1 | Gran útil | sing. |

(b) Controlling gender when translating "The interested *parties*" (left) and "*Questions* relating to information" (right) to Spanish.

| $\alpha$ | Translation | Gen | $\alpha$ | Translation | Gen |
|---|---|---|---|---|---|
| -0.5, -0.25 | Los partidos interados | ms. | -1 | Temas relativos a la información | ms. |
| 0, 0.25 | Las partes interesadas | fm. | -0.5, 0, 0.5 | Cuestiones relativas a la información | fm. |

(c) Controlling tense when translating "The committee *supported* the efforts of the authorities".

| | $\alpha$ | Translation | Tense |
|---|---|---|---|
| Arabic | –/+10 | وأيدت\وتؤيد اللجنة {جهود\الجهود التي تبذلها} السلطات | past/present |
| French | –/-20 | Le Comité a appuyé/appuie les efforts des autorités | past/present |
| Spanish | –/-3/0 | El Comité apoyó/apoyaba/apoya los esfuerzos de las autoridades | past/impf./present |
| Russian | –/-1 | Комитет поддержал/поддерживает усилия властей | past/present |
| Chinese | –/-50 | 委员会 支持 当局 的 努力 / 委员会 正在 支持 当局 的 努力 | untensed/present |

**Number** Table 4a shows translation control results for a number neuron from an English-Spanish model, which activates negatively/positively on plural/singular nouns. The translation changes from plural to singular as we increase the modification $\alpha$. We notice that using too high $\alpha$ values yields nonsense translations, but with correct number: transitioning from the plural adjective *particulares* ("particular") to the singular adjective *útil* ("useful"). In between, we see a nice transition between plural and singular translations. Interestingly, the translations exhibit correct agreement between the modified noun and its adjectives and determines, e.g., *Las partes interesadas* vs. *La parte interesada*. This is probably due to the strong language model in the decoder.

**Gender** Table 4b shows examples of controlling gender translation for a gender neuron from the same model, which activates negatively/positively on masculine/feminine nouns. The translations of "parties" and "questions" change from masculine to feminine synonyms as we increase the modification $\alpha$. Generally, we found it difficult to control gender, as also suggested by the relatively low success rate.

**Tense** Table 4c shows examples of controlling tense when translating from English to five target languages. In all language pairs, we are able to change the translation from past to present by modifying the activation of the tense neurons from the previous section (Table 3). Occasionally, modifying the activation on a single word leads to a change in phrasing; in Arabic the translation changes to "the efforts that the authorities invest". In Spanish, we find a transition from past (*apoyó*) to imperfect (*apoyaba*) to present (*apoya*). Interestingly, in Chinese, we had to use a fairly large $\alpha$ value (in absolute terms), consistent with the fact that tense is not usually marked in Chinese. In fact, our modification generates a Chinese expression (正在) that is used to express an action in progress, similar to English "-ing", resulting in the meaning "is supporting".

## 7 CONCLUSION

Neural machine translation models learn vector representations that contain linguistic information while being trained solely on example translations. In this work, we developed unsupervised methods for finding important neurons in NMT, and evaluated how these neurons impact translation

quality. We analyzed several linguistic properties that are captured by individual neurons using quantitative prediction tasks and qualitative visualizations. We also designed a protocol for controlling translations by modifying neurons that capture desired properties.

Our analysis can be extended to other NMT components (e.g. the decoder) and architectures (Gehring et al., 2017; Vaswani et al., 2017), as well as other tasks. We believe that more work should be done to analyze the spectrum of localized vs. distributed information in neural language representations. We would also like to expand the translation control experiments to other architectures and components (e.g. the decoder), and to develop more sophisticated ways to control translation output, for example by modifying representations in variational NMT architectures (Zhang et al., 2016; Su et al., 2018). Our code is publicly available as part of the NeuroX toolkit (Dalvi et al., 2019b).[7]

## ACKNOWLEDGEMENTS

This research was carried out in collaboration between the HBKU Qatar Computing Research Institute (QCRI) and the MIT Computer Science and Artificial Intelligence Laboratory (CSAIL). Y.B. is also supported by the Harvard Mind, Brain, and Behavior Initiative (MBB).

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

# A  ADDITIONAL RESULTS AND VISUALIZATIONS

## A.1  NOUN PHRASE SEGMENTATION

Table 5 shows the top neurons from each network by accuracy when classifying interior, exterior, or beginning of a noun phrase. The annotations were obtained with Spacy. We found high-scoring neurons (60-80% accuracy) in every network. Many of these neurons were ranked highly by the `MaxCorr` ranking methods. In contrast, other correlation methods did not rank such neurons very highly. Thus there is correspondence between a high rank by our intrinsic unsupervised measure `MaxCorr` and the neuron's capacity to predict external annotation.

Table 5: Top neuron from each network by accuracy for classifying interior, exterior, or beginning of a noun phrase, as well as ranking of these neurons by our intrinsic unsupervised measures.

| | | Rank | | |
| Neuron | Accuracy | MaxCorr | MinCorr | LinReg |
|---|---|---|---|---|
| en-es-1:221 | 0.78 | 9 | 53 | 58 |
| en-es-2:158 | 0.76 | 12 | 60 | 65 |
| en-es-3:281 | 0.72 | 25 | 40 | 111 |
| en-fr-1:111 | 0.76 | 13 | 63 | 123 |
| en-fr-2:85 | 0.74 | 33 | 46 | 88 |
| en-fr-3:481 | 0.75 | 14 | 66 | 135 |
| en-ar-1:168 | 0.71 | 17 | 59 | 49 |
| en-ar-2:190 | 0.76 | 48 | 81 | 92 |
| en-ar-3:383 | 0.70 | 16 | 80 | 77 |
| en-ru-1:38 | 0.66 | 36 | 55 | 159 |
| en-ru-2:130 | 0.67 | 35 | 65 | 136 |
| en-ru-3:78 | 0.66 | 160 | 108 | 124 |
| en-zh-1:427 | 0.64 | 23 | 76 | 240 |
| en-zh-2:199 | 0.65 | 187 | 218 | 233 |
| en-zh-3:28 | 0.67 | 65 | 33 | 44 |

## A.2  CONTROLLING TRANSLATIONS

We provide additional translation control results. Table 6 shows the tense results using the best modification value from Figure 6a. We report the number of times the source word was aligned to a target word which is past or present, or to multiple words that include both or neither of these tenses. The success rate is the percentage of cases where the word was aligned to a target word with the desired tense. By modifying the activation of only one neuron (the most predictive one), we were able to change the translation from past to present in 67% of the times and vice-versa in 49% of the times. In many other cases, the tense was erased, that is, the modified source word was not aligned to any tensed word, which is a partial success.

Table 6: Results for controlling tense.

| From \ To | Past | Present | Both | Neither | Success Rate |
|---|---|---|---|---|---|
| Past | 85 | 820 | 9 | 311 | 67% |
| Present | 1586 | 256 | 30 | 1363 | 49% |

### A.3 Visualizations

Here we provide additional visualizations of neurons capturing different linguistic properties.

**Noun phrases** We visualize the top scoring neuron (79%) from an English-Spanish model in Figure 7. Notice how the neuron activates positively (red color) on the first word in the noun phrases, but negatively (blue color) on the rest of the noun phrase (e.g. "Regional" in "Regional Service Centre").

Figure 7: Visualization of a neuron from an English-Spanish model that activates positively (red color) on the first word in the noun phrase and negatively (blue) on the following words.

**Dates and Numbers** Figure 8 shows activations of neurons capturing dates and numbers. These neurons were ranked highly (top 30) by `MaxCorr` when ranking an English-Arabic model trained with charCNN representations. We note that access to character information leads to many neurons capturing sub-word information such as years (4-digit numbers). The first neuron is especially sensitive to month names ("May", "April"). The second neuron is an approximate year-detector: it is sensitive to years ("2015") as well as other tokens with four digits ("7439th", "10.15").

(a) Month neuron

(b) Approximate "year" neuron

Figure 8: Neurons capturing dates and numbers.

**List items** Figure 9 shows an interesting case of a neuron that is sensitive to the appearance of two properties simultaneously: position in the beginning of the sentence and number format. Notice that it activates strongly (negatively) on numbers when they open a sentence but not in the middle of the sentence. Conversely, it does not activate strongly on non-number words that open a sentence. This neuron aims to capture patterns of opening list items.

### A.4 Results on models from different epochs

In the main experiments in the paper, we have used models trained on different parts of the training data, as well as on different language pairs. However, our methodology can be applied to any collection of models that we think should exhibit correlations in their neurons. We have verified that this approach works with model checkpoints from different epochs of the same training run. Concretely, we computed `MaxCorr` scores for the last checkpoint in two models—English-Spanish and English-Arabic—when comparing to other checkpoints. In both cases, we found highly correlated neurons across checkpoints, especially in the last few checkpoints. We also observed that erasing the top neurons hurt translation performance more than erasing the bottom neurons, similar to the findings in Section 5.1. Moreover, we noticed a significant overlap between the top ranked neurons in this case and the ones found by correlating with other models, as in the rest of the paper. In particular, for the English-Spanish model, we found that 8 out of 10 and 34 out of 50 top ranked neurons are the same in these two rankings. For the English-Arabic model, we found a similar behavior (7 out of 10 and 33 out of 50 top ranked neurons are the same). This indicates that our method may be applied to different checkpoints as well.

7439th meeting , held on 11 May 2015 .

ISIL itself has published videos depicting people being subjected to a range of abhorrent punishments , including stoning , being pushed-off buildings , decapitation and crucifixion .

UNICEF disbursed emergency cash assistance to tens of thousands of displaced families in camps and UNHCR distributed cash assistance to vulnerable families which had been internally displaced .

31 . Recognizes the important contribution of the African Peer Review Mechanism since its inception in improving governance and supporting socioeconomic development in African countries , and recalls in this regard the high-level panel discussion held on 21 October 2013 on Africa 's innovation in governance through 10 years of the African Peer Review Mechanism , organized during the sixty-eighth session of the General Assembly to commemorate the tenth anniversary of the Mechanism ;

Spreads between sovereign bonds in Germany and those in other countries were relatively unaffected by political and market uncertainties concerning Greece in late 2014 and early 2015 .

The removal of the floor with respect to the euro was accompanied by a further move into negative territory of the interest rate on sight deposit account balances to -0.75 per cent in order to reduce appreciating pressures and the resulting tightening of monetary conditions .

To be held on Thursday , 2 April 2015 , at 10.15 a.m.

Upon instruction from my Government , I have the honour to attach herewith a list containing the names of 96 Syrian civilians , including 41 children , killed in Aleppo by terrorist groups during the period from 13 April 2015 to 7 May 2015 ( see annex ) .

12 . Reiterates that individuals and entities determined by the Committee to have violated the provisions of resolution 1970 ( 2011 ) , including the arms embargo , or assisted others in doing so , are subject to designation , and notes that this includes those who assist in the violation of the assets freeze and travel ban in resolution 1970 ( 2011 ) ;

23 . Supports the efforts of the Libyan authorities to recover funds misappropriated under the Qadhafi regime and , in this regard , encourages the Libyan authorities and Member States that have frozen assets pursuant to resolutions 1970 ( 2011 ) and 1973 ( 2011 ) as modified by resolution 2009 ( 2011 ) to consult with each other regarding claims of misappropriated funds and related issues of ownership ;

Figure 9: A neuron that activates on numbers in the beginning of sentences. The first 10 sentences in the test set are shown.

## B   A CATALOG OF TOP RANKED NEURONS

In order to illustrate the range of linguistic phenomena captures by individual neurons, we provide here a list of the top 20 neurons (or projected directions, in the case of SVCCA) found by each of our methods, for an example English-Spanish model. For each neuron, we give the percentage of variance that is eliminated by conditioning on position in the sentence or identity of the current token. We also comment on what properties each neuron appears to capture, based on visualizations of neuron activation. Where possible, we give $F_1$ scores of a GMM model predicting certain properties such as detecting noun phrase segmentation, parenthetical phrases, adjectives, and plural nouns. Annotations are obtained with Spacy: `https://spacy.io`.

Table 7: Top 20 ranked neurons by `MaxCorr`.

| Neuron | Position | Token | Comments |
|---|---|---|---|
| 464 | 92% | 10% | Position. |
| 342 | 88% | 7.9% | Position. |
| 260 | 0.71% | 94% | Conjunctions: "and", "or", "well", "addition". |
| 49 | 11% | 6.1% | Activates for several words after "and" or "or". |
| 124 | 77% | 48% | Position. |
| 394 | 0.38% | 22% | Noun phrase segmentation. 13th-best F1-score (0.41) for finding interiors of noun phrases. 15th-best IOB accuracy (0.62). |
| 228 | 0.38% | 96% | Unknown: "'s", "the", "this", "on", "that". |
| 133 | 0.14% | 87% | Adjective detector. Best F1-score (0.56) for finding adjectives. |
| 221 | 1% | 30% | Noun phrase segmentation. Best F1-score for finding interiors (0.72) and for finding beginnings (0.63). Best IOB accuracy (0.78). |
| 90 | 0.49% | 28% | Noun phrase segmentation. Second-best F1-score (0.56) for finding beginnings of noun phrases. Second-best IOB accuracy (0.72). |
| 383 | 67% | 6.5% | Position. |
| 494 | 3.5% | 96% | Punctuation/conjunctions: ",", ";", "Also", "also", "well". |
| 120 | 0.094% | 84% | Plural noun detector. Best F1-score (0.85) for retrieving plural nouns. |
| 269 | 0.1% | 80% | Spanish noun gender detector. Very positive for "islands", "activities", "measures" – feminine. Very negative for "states", "principles", "aspects" – masculine. |
| 232 | 0.63% | 31% | Parentheses. Best F1-score (0.66) for retrieving tokens inside parentheses/quotes/brackets. |
| 332 | 0.13% | 83% | Unknown. |
| 324 | 0.18% | 81% | Unknown. |
| 210 | 0.61% | 45% | Date detector. Third-best F1-score (0.36) for retrieving tokens inside dates. |
| 339 | 0.48% | 39% | Activates for a verb and also surrounding inflection words/auxiliary verbs. |
| 139 | 0.86% | 93% | Punctuation/conjunctions: ",", ".", "–". |

Table 8: Top 20 ranked neurons by `MinCorr`.

| Neuron | Position | Token | Comments |
|---|---|---|---|
| 342 | 88% | 7.9% | Position. |
| 464 | 92% | 10% | Position. |
| 260 | 0.71% | 94% | Conjunctions: "and", "or", "well", "addition". |
| 383 | 67% | 6.5% | Position. |
| 250 | 63% | 6.8% | Position. |
| 124 | 77% | 48% | Position. |
| 485 | 64% | 10% | Position. |
| 480 | 70% | 12% | Position. |
| 154 | 63% | 15% | Position. |
| 139 | 0.86% | 93% | Punctuation/conjunctions: ",", ".", "–", "alia". |
| 20 | 60% | 9.2% | Position. |
| 494 | 3.5% | 96% | Punctuation/conjunctions: ",", ";", "also", "well". |
| 199 | 67% | 6% | Position. |
| 126 | 42% | 9.4% | Unknown. |
| 348 | 50% | 13% | Position. |
| 46 | 48% | 8.6% | Unknown. |
| 196 | 60% | 8.5% | Position. |
| 367 | 0.44% | 89% | Prepositions: "of", "or", "United", "de". |
| 186 | 1.6% | 69% | Conjunctions: "also", "therefore", "thus", "alia". |
| 244 | 54% | 15% | Position. |

Table 9: Top 20 ranked neurons by `LinReg`.

| Neuron | Position | Token | Comments |
|---|---|---|---|
| 464 | 92% | 10% | Position. |
| 260 | 0.71% | 94% | Conjunctions: "and", "or", "well", "addition". |
| 139 | 0.86% | 93% | Punctuation/conjunctions: ",", ".", "–", "alia". |
| 494 | 3.5% | 96% | Punctuation/conjunctions: ",", ";", "also", "well". |
| 342 | 88% | 7.9% | Position. |
| 228 | 0.38% | 96% | Possibly determiners: ""s", "the", "this", "on", "that". |
| 317 | 1.5% | 83% | Indefinite determiners: "(", "one", "a", "an". |
| 367 | 0.44% | 89% | Prepositions. "of", "for", "United", "de", "from", "by", "about". |
| 106 | 0.25% | 92% | Possibly determiners: "that", "this", "which", "the". |
| 383 | 67% | 6.5% | Position. |
| 485 | 64% | 10% | Position. |
| 186 | 1.6% | 69% | Conjunctions. "also", "therefore", "thus", "alia". |
| 272 | 2% | 73% | Tokens that mean "in other words": "(", "namely", "i.e.", "see", "or". |
| 124 | 77% | 48% | Position. |
| 480 | 70% | 12% | Position. |
| 187 | 1.1% | 87% | Unknown: "them", "well", "be", "would", "remain". |
| 201 | 0.14% | 73% | Tokens that mean "regarding": "on", "in", "throughout", "concerning", "regarding". |
| 67 | 0.27% | 71% | Unknown: "united", "'s", "by", "made", "from". |
| 154 | 63% | 17% | Position. |
| 72 | 0.32% | 89% | Verbs suggesting equivalence: "is", "was", "are", "become", "constitute", "represent". |

Table 10: Top 20 ranked directions by SVCCA.

| Position | Token | Comments |
|---|---|---|
| 86% | 26% | Position |
| 1.6% | 90% | Detects "the". |
| 7.5% | 85% | Conjunctions: "and", "well", "or". |
| 20% | 79% | Determiners: "the", "this", "these", "those". |
| 1.1% | 89% | Possibly conjunctions: negative for "and", "or", "nor", positive for "been", "into", "will". |
| 10% | 76% | Punctuation/conjunctions: positive for ",", ";", "." "–", negative for "and". |
| 30% | 57% | Possibly verbs: "been", "will", "be", "shall". |
| 24% | 55% | Possibly date detector. |
| 23% | 60% | Possibly adjective detector. |
| 18% | 63% | Unknown. |
| 4.5% | 88% | Punctuation: ".", ",", ";" |
| 9.8% | 69% | Forms of "to be": "is", "will", "shall", "would", "are". |
| 1.7% | 77% | Combined dates/prepositions/parentheses: negative for "in", "at", ".", positive for dates and in quotes/parentheses/brackets. Noisy. |
| 16% | 25% | Activates for a few words after "and". |
| 14% | 63% | Possibly plural noun detector. |
| 0.8% | 73% | Spanish noun gender detector. |
| 11% | 61% | Possibly singular noun detector. |
| 13% | 58% | Possibly possessives: "its", "his", "their". |
| 1.4% | 73% | Spanish noun gender detector. |
| 5.6% | 53% | Unknown. |

