# OpenReview forum: "Identifying and Controlling Important Neurons in Neural Machine Translation"
_ICLR.cc/2019/Conference_

### Official Review · AnonReviewer3 · 2018-10-22
**Well-written paper applying a method for finding individual influential neurons to MT, but insight is ultimately limited**

**Rating:** 6
**Confidence:** 4

**Review:**

The authors propose a number of methods to identify individual important neurons in a machine translation system. The crucial assumption, drawn from the computer vision literature, is that important neurons are going to be correlated across related models (e.g. models that are trained on different subsets of the data). This hypothesis is validated to some extent: erasing the neurons that scored highly on these measures reduced BLEU score substantially. However, it turns out that most of the activation of the important neurons can be explained using sentence position. Supervised classification experiments on the important neurons revealed neurons that tracked properties such as the span of parentheses or word classes (e.g., auxiliary verbs, plural nouns, etc).

Strengths:
* The paper is very well written and provides solid intuitions for the methods proposed.
* The methods seem promising, and the degree of localist representation is striking.
* The methods may be able to address the question of *how* localist the representations are (though no numerical measure of localism is proposed).
* There is a correlation between the neuron importance metrics proposed in the paper and the effect on BLEU score of erasing those neurons from the network (of course, it’s not clear what particular linguistic properties are affected by this erasure - the decrease BLEU may reflect inability to track specific word tokens more than any higher-level linguistic property).

Weaknesses:
* It wasn't clear to me why the neurons that track particular properties (e.g., being inside a parentheses) couldn't be identified using a supervised classifier to begin with, without first identifying "important" neurons using the unsupervised methods proposed in the paper. The unsupervised methods do show their strength in the more exploratory visualization-based analyses -- as the authors point out (bottom of p. 6), the neuron that activates on numbers but only at the beginning of the sentence does not correspond to a plausible a-priori hypothesis. Still, most of the insight in the paper seems to be derived from the supervised experiments.
* The particular linguistic properties that are being investigated in the classification experiments are fairly limited. Are there neurons that track syntactic dependencies, for example?
* I wasn't sure how the GMMs (Gaussian mixture models) for predicting linguistic properties from neuron activations were set up.
* It's nice to see that individual neurons function as knobs that can change the gender or tense of the output (with varying accuracy). At the same time, I was unable to follow the authors' argument that this technique could be used to reduce gender bias in MT.
* I wasn't sure what insight was gained from the SVCCA analyses -- this method seems to be a bit of a distraction given the general focus on localist vs. distributed representation. In general, I didn’t come away with an understanding of the pros and cons of each of the methods.

---

> ### Author Response · Authors · 2018-11-19
> **Response to Reviewer 3: part 1/2**
>
> Thank you for your useful feedback and helpful comments. We are glad that you found our methods promising with good intuitions. We would like to clarify a few points based on your comments.
>
> 1. “Most of the activations of the important neurons can be explained using sentence position”
> It is true that many top ranked neurons capture sentence position, especially in the MinCorr method (Table 1). However, other methods reveal important neurons that do not capture position: only 3/10 top ranked neurons by MaxCorr and LinReg are position neurons, and only one top ranked SVCCA direction captures position. These top neurons often capture linguistic properties like morphological features, punctuations, word classes, etc., as analyzed in Section 5.3 and in appendices A and C. We will make this point clearer in the next revision.
>
> 2. “The methods may be able to address the question of *how* localist the representations are (though no numerical measure of localism is proposed)”
> We also believe that our methods can shed light on the question of how localist the representations are. We would love to try out any suggestions for numerical measures of localism.
>
> 3.”why the neurons that track particular properties couldn't be identified using a supervised classifier to begin with” and “most of the insight in the paper seems to be derived from supervised experiments”
> The advantage of unsupervised methods is that they are not constrained by available supervision in the form of linguistic annotations. Our working process involved visualizations of the important neurons, which led to forming hypotheses about their function. In order to validate our interpretations, we designed supervised experiments whenever we could. While this may give the impression that most insights are derived from the supervised experiments, in practice it would have been difficult to choose specific properties to target without the unsupervised methods + visualizations.
> In addition, we have found properties that do not correspond to plausible a priori hypotheses. The neuron detecting item numbers which you mention is one such case. We also found a neuron that activates positively on the first word in a noun phrase and negatively in the rest of the phrase (Figure 5). Other properties that may not be expected to emerge include year and month neurons (Figure 6), a neuron that activates on verbs and their surrounding words (Table 7), and neurons that capture both punctuation and conjunctions (Tables 6+7; note that this would not be captured by standard part-of-speech tag sets).
> We will improve our presentation according to your feedback.
>
> 4. “are there neurons that track syntactic dependencies, for example?”
> In this work we focused mainly on word-level properties. However, we did investigate parentheses, which require longer-range context, and also noun phrases. Still, we agree that it would be interesting to consider more compositional properties such as dependencies or phrase structure.
>
> 5. “how the GMMs … were set up”
> The GMMs were set up to predict a property from a neuron activation. The number of mixture was chosen as the number of different classes in the prediction task. For instance, for finding parenthesis neurons we used two classes (inside or outside of parentheses/quotes/brackets). We estimated the parameters of the GMMs using the mean and variance of the neuron activation conditioned on each class. We tested the resulting model to see how well it predicts the tag from the neuron activation by computing the posterior probability of each class given an activation using Bayes' rule, and taking the argmax. We will provide more details on the GMM in the next revision.
>
> 6. “argument that this technique could be used to reduce gender bias in MT”
> Our reasoning in the control experiments is that some information about sensitive attributes like gender may be available from other sources, such as metadata. If we know that an entity has a specific gender (say, feminine), but that gender is unmarked in the English language (as in the word “doctor”), then we may encourage the system to output a translation with the correct gender by modifying gender neurons. This is a kind of soft constraint that we may add to the system. We will improve the motivation in the next revision.

---

> ### Author Response · Authors · 2018-11-19
> **Response to Reviewer 3: part 2/2**
>
>
> 7. “what insight was gained from the SVCCA analyses” and “the pros and cons of each of the methods”
> Thank you for the feedback. We will improve the discussion in the next revision accordingly.
> The different methods aim to analyze representations at different levels of localism/distributivity. In particular, while MaxCorr and MinCorr target pairwise neuron correlations, LinReg searches for information that’s distributed in the whole representation in one network, but localized in a different network. SVCCA tries to find a middle ground by projecting representations to a lower-dimensional space and then computing correlations.
> Some of the insights are discussed in Section 5.2, where we observe that the more distributed methods (LinReg and SVCCA) give much importance to identifying specific tokens. This means that information about token identity is distributed among many neurons. The fact that MinCorr cares a lot about position suggests that this kind of information is captured in multiple models in a similar way. Table 10 in the appendix shows that in addition to detecting specific tokens, SVCCA directions may also capture some classes like adjectives and verbs.

---

### Official Review · AnonReviewer1 · 2018-10-30
**This paper presents unsupervised methods for ranking neurons in machine translation. Important neurons are thus identified and used to control the MT output.**

**Rating:** 10
**Confidence:** 3

**Review:**

Strengths:
- even though the methods for detecting important neurons are not novel (as also stated in the paper), their application to MT is novel
- the presentation is very clear
- the choice of methods is well argued and justified
- the experiments are well executed and analysed
- thorough and varied analysis of the experimental findings

I recommend this paper for the best paper award.

---

> ### Author Response · Authors · 2018-11-19
> **Response to Reviewer 1**
>
> Thank you for your very positive review. We are glad that you found the choice of methods justified, and the experiments and analysis thorough and well executed.

---

### Official Review · AnonReviewer2 · 2018-11-02
**Interesting analysis of the contributions of different neurons in NMT**

**Rating:** 7
**Confidence:** 3

**Review:**

This paper presents unsupervised approaches to discover import neurons in
neural machine translation systems. Some linguistic properties controlled by the
discovered neurons are discussed and analyzed.

Strengths:

The paper is well-written and provides valuable information to understand the
behaviour of neural machine translation models.

The ability to control characteristics (such as gender) without training
specialized models is promising, even if the results are not good enough for
immediate use. It would be interesting to see whether controlling neurons
in the decoder would be more effective.

Weaknesses:

Multiple NMT systems are necessary to discover important neurons. The authors
mention that it would be possible to use different checkpoints from a single
model, but don't evaluate how well this would work.

The findings in this paper do not lead to immediate translation performance
improvements.

Questions and other remarks:

In Table 4a, why are there 2 results for "-0.25, -0.125, 0"?

In section 4.3 (Tense), it may be worthwhile to mention that the neuron is
highly activated on the word "Spreads", even if it acts as a noun in this
specific sentence.

Bottom of p. 6: "Our supervised methods" -> "Our unsupervised methods"

To control properties, could SVCCA directions or coefficients be manipulated?

Some parentheses around citations are missing or misplaced.

---

> ### Author Response · Authors · 2018-11-19
> **Response to Reviewer 2**
>
> Thank you for your positive and constructive feedback. We are happy that you find our analysis interesting and valuable for understanding the behavior of neural MT models. We answer specific comments below.
>
> 1.  “Controlling neurons in the decoder”
> We are also interested in expanding the controlling experiments, both to other properties and to the decoder side, and intend to do so in the future. We will mention this as potential future work.
>
> 2. “use different checkpoints from a single model”
> Thank you for bringing this point up. We have compared all checkpoints from a couple of our models and found highly correlated neurons, especially when correlating later checkpoints. This makes sense as the model converges to a solution. We verified that these top correlated neurons are important for the model performance via an erasure experiment similar to Section 5.1.
> Moreover, the top ranked neurons when comparing the last checkpoint to earlier ones are very similar to the ones found when comparing this last checkpoint to different models, including models trained with different target languages. In particular, for the English-Spanish model, we found that 8 out of 10 and 34 out of 50 top ranked neurons are the same in these two rankings. For the English-Arabic model, we found a similar behavior (7 out of 10 and 33 out of 50 top ranked neurons are the same). This indicates that our method may be applied to different checkpoints as well. We will add these results in the next revision.
>
> 3. “The findings in this paper do not lead to immediate translation performance improvements”
> This is correct. Beyond the scientific value in illuminating how NMT models work, we would also like to mention several potential ideas for improving the systems. First, our experiments for controlling specific characteristics may help mitigate model bias by identifying neurons responsible for sensitive attributes such as gender or politeness. For instance, we might have external knowledge of the gender of person mentioned by an ambiguous profession or title in the source language (e.g., doctor), and may want to encourage the translation to be of the correct gender in the target language (as the Turkish example in Section 6 illustrates). More generally, by identifying neurons that are responsible for common mistakes we may be able to improve the system through similar control experiments. Other directions for improving NMT systems are doing model compression by removing unimportant neurons and guiding neural architecture search by tracking important neurons.
>
> 4. “Table 4a, two results”
> Thank you. We have fixed this typo.
>
> 5. Tense neuron activating on “Spreads”
> Indeed, this is a “mistake” of the neuron. We will mention this.
>
> 6. "Our supervised methods" → "Our unsupervised methods"
> Fixed. Thank you.
>
> 7. “Could SVCCA directions be manipulated”
> This is difficult to do, as it requires changing all the dimensions, rather than a small number of dimensions, and we’ve noticed that modifying more dimensions leads to performance degradation. Moreover, SVCCA directions mostly detect specific tokens rather than a linguistic property (see Table 1) so controlling is not very intuitive in this case.
>
> 8. Missing or misplaced parentheses
> We have fixed those. Thank you.

---

### Author Response · Authors · 2018-11-22
**Updated version incorporating reviewers' comments**

We have uploaded a revised version incorporating the reviewers's helpful comments. This is the list of changes:

1. Added appendix A.4 with results with different model checkpoints and a footnote referring to the appendix in Section 4.
2. Added to the Conclusion a potential future work on controlling translations by modifying the decoder.
3. Mentioned the tense neuron’s mistake on "Spreads" in Section 5.3, tense paragraph.
4. Slightly modified sections 5.2 and 5.3 to emphasize that position is captured by top neurons in some of the unsupervised methods, but not in all.
5. Clarified that the supervised experiments are used for verifying interpretations, rather than constraining the analysis to a given property that may not be known a priori (Section 3.2, supervised verification).
6. Added more details on the GMM setup in Section 3.2, supervised verification.
7. Improved the motivation for using translation control for mitigating model bias in the beginning of Section 6.
8. Fixed a number of typos and formatting issues.

---

### Public Comment · (anonymous) · 2019-02-14
**Interesting analysis, link to another paper also controlling by neuron modifications, but in language models**

Very interesting paper, thanks!

In particular, I read with great interest your protocol to control translations and wanted to point you to a recently published paper that also performs such 'intervention' experiments, but on language models:

http://aclweb.org/anthology/W18-5426

Hope to read more about follow up studies soon.

---

### Meta-Review · Area_Chair1 · 2018-12-13
**An insightful paper presenting analyses of recurrent machine translation models**

**Confidence:** 5
**Recommendation:** Accept (Poster)

**Metareview:**

Strong points:

-- Interesting, fairly systematic and novel analyses of recurrent NMT models, revealing individual neurons responsible for specific type of information (e.g., verb tense or gender)

-- Interesting experiments showing how these neurons can be used to manipulate translations in specific ways (e.g., specifying the gender for a pronoun when the source sentence does not reveal it)

-- The paper is well written

Weak points

-- Nothing serious (e.g., maybe interesting to test across multiple runs how stable these findings are).

There is a consensus among the reviewers that this is a strong paper and should be accepted.